# OpenReview forum: "Chain-of-Thought Reasoning In The Wild Is Not Always Faithful"
_ICLR.cc/2026/Conference — Submitted to ICLR 2026_

### Official Review · Reviewer_iSVc · 2025-10-29

**Soundness:** 3
**Presentation:** 3
**Contribution:** 3
**Rating:** 6
**Confidence:** 3

**Summary:**

This paper systematically investigates the problem of unfaithfulness in CoT reasoning within LLMs. The authors identify two previously underexplored phenomena:

1. Implicit Post-Hoc Rationalization — when faced with logically exclusive problems, the model generates seemingly plausible yet mutually contradictory reasoning chains to justify an underlying default answer bias.
2. Unfaithful Illogical Shortcuts — in mathematical reasoning, the model arrives at correct answers through “shortcut” or erroneous reasoning steps that conceal logical inconsistencies within its CoT process.

Extensive experiments were conducted on 15 LLMs using a contrastive dataset of 4,834 questions, demonstrating the prevalence of unfaithful reasoning behaviors in mathematical tasks.

**Strengths:**

1. The paper provides a systematic analysis of the faithfulness problem in CoT reasoning, addressing a research question of significant value.
2. Extensive experiments demonstrate that such faithfulness issues are prevalent across most models, including those equipped with reasoning capabilities, offering valuable insights and guidance for future large language model research.

**Weaknesses:**

1. Although the paper identifies implicit biases, it does not uncover the underlying triggering mechanisms within the Transformer architecture. An internal analysis of such mechanisms would make the study more convincing and interpretable.
2. The work lacks rigorous significance testing or bootstrap confidence intervals.
3. The task coverage is limited — the dataset focuses mainly on comparative and mathematical reasoning, without including more complex scenarios such as multi-hop or conversational reasoning.
4. The definition of faithfulness remains ambiguous, as it still conflates observable consistency with cognitive consistency; the theoretical boundary between the two requires clearer delineation.

**Questions:**

1. Can the causal contribution of internal bias activation to the unfaithfulness rate be quantitatively measured? Could this be verified using logit-lens analysis or attention probing?
2. In PutnamBench, is the phenomenon of Unfaithful Illogical Shortcuts significantly correlated with training data contamination?
3. Could the proposed Consistency-with-Reversal regularizer be integrated into RLHF or DPO pipelines for empirical validation?
4. Can the framework be extended to open-ended reasoning tasks such as multi-hop question answering or commonsense reasoning?

---

> ### Author Response · Authors · 2025-11-24
>
> Dear Reviewer iSVc,
>
> Thank you for the constructive review. We respond to your main points below. If these clarifications and the changes made to our revised version address your concerns, we would be grateful if you could update your score.
>
> ## Weakness 1: Lack of internal analysis for mechanisms
>
> We totally agree that internal mechanistic analysis of how and why models are unfaithful is extraordinarily valuable. But we did not want to have scope creep with our work, doing too many things at once. Therefore, we think that our work is an extremely solid contribution to creating a robust dataset for unfaithfulness, and that this will provide a great basis for future work to do unfaithfulness internal analysis.
>
> That said, we already provide some internal evidence. In Appendix F we train linear probes on residual activations of Llama 3.3 70B It and show that they can predict a large fraction of the template level Yes/No bias at specific layers and tokens, before the written reasoning starts. This suggests that the biases are encoded in internal representations, not only in surface wording.
>
> ## Weakness 2: Lack of confidence Intervals
>
> We've added 95% bootstrap confidence intervals to all the plots in the revised version of the paper (submitted with this rebuttal). We appreciate you bringing this up, since it strengthens the statistical rigour of our analysis and makes the robustness of our findings clearer.
>
> ## Weakness 3 & Question 4: Limited task coverage
>
> IPHR is built from 29 subsets of the World Model dataset, with comparative questions over books, movies, people, geographic entities, and so on. Many questions require short multi step reasoning, for example combining knowledge about two locations and then applying a comparison rule. We now clarify this in the text.
>
> We agree that we do not cover open ended multi hop QA or conversational settings, where answers are long and less structured. Our framework can in principle be applied there by constructing pairs of prompts that share a factual base but flip the correct label, then checking for inconsistent answers and CoTs across the pair. The main challenge is evaluation of long generations, which would need new automatic metrics and likely human checks. We add a paragraph in the limitations that explains these challenges and lists open ended multi hop and conversational tasks as future extensions.
>
> ## Weakness 4: Better definition of unfaithfulness
>
> We agree that the distinction between observable consistency and "cognitive" consistency deserves clearer wording. Our experiments study what we now call behavioural faithfulness: whether the written CoT is systematically consistent with the model's behaviour across paired prompts that differ only in a controlled way.
>
> There are many cases where the model's answer changes between a question and its logical reverse, even though the visible reasoning does not mention the feature that is flipped (for example, the order of entities). This shows that the written CoT cannot fully explain the behaviour. However, we do not recover the exact internal computation that caused the discrepancy. In the revision we use the term behavioural faithfulness for our metrics, clarify that we test a necessary but not sufficient condition for cognitive faithfulness, and discuss this boundary in the related work and limitations sections.
>
> ## Question 1: Causal role of internal bias activations
>
> Building on Appendix F, we see a simple causal test as natural future work. The idea is to treat the probe directions that predict template bias as candidate bias directions in activation space, and at inference time steer activations along or against these directions to measure how the Yes/No bias and unfaithful pair rate change. We now mention this plan explicitly as a future direction.
>
> ## Question 2: Training data contamination on PutnamBench
>
> Section 3.1.1 already tests the contamination hypothesis. We run our full pipeline on problems from the 2024 Putnam exam, which was released shortly after the training cut off of all models we study. The model still produces a substantial number of unfaithful illogical shortcuts on these unseen problems.
>
> ## Question 3: Consistency with reversal regulariser and RLHF / DPO
>
> Our consistency with reversal idea is to define, for each pair of logically opposite questions, a loss that rewards answer pairs that are both correct and mutually consistent and penalises inconsistent pairs. This signal can be added on top of supervised fine tuning or DPO by treating consistent pairs as preferred samples, and in RLHF by giving higher rewards to rollouts that remain consistent across reversed prompts. We expand the description of this idea as a concrete training signal to explore in future work.

---

### Official Review · Reviewer_7nJy · 2025-10-31

**Soundness:** 2
**Presentation:** 4
**Contribution:** 3
**Rating:** 4
**Confidence:** 4

**Summary:**

The authors present a study of chain-of-thought (CoT) unfaithfulness, where LLMs produce answers that do not necessarily reflect the reasoning in the predicted CoT. They conduct their study from two angles: one exploring the model's behavior when answering natural language questions containing anti-symmetric relations (e.g., is X less than Y? vs is Y less than X?). The second angle of their study is complex math word problem solving, where the model may produce invalid steps in order to justify a predetermined response.

**Strengths:**

- The authors propose an interesting study to quantitatively measure the unfaithfulness of the model's CoT.
 - They conduct experiments on a wide range of models, including both thinking and non-thinking models, from disparate model families.
 - The paper is well-written and easy to understand.

**Weaknesses:**

- There are a number of confounding factors that may explain some of the observed discrepancies in the model (see below for further details).
 - A validation study is missing for the LLM-as-a-judge approach in evaluation.

**Questions:**

The example of whether the Ajay River is to the south of Salar de Arizaro (or vice versa) highlights another possible explanation for the model's unchanged response (no): Perhaps there are more than two possible answers. That is, there is at least one other answer in addition to "Ajay River is located to the south of Salar de Arizaro" and "Salar de Arizaro is located to the south of the Ajay River". This is because "to the south of" may not be a strictly anti-symmetric relation (i.e., a relation `r` is anti-symmetric if for any `x`, `y`, `r(x,y)` if and only if not `r(y,x)`). For example, this confounding factor arises in examples where r(x,y) and r(y,x) are both false (or both true), such as in "is 2i larger than sqrt(-4)?" It can also arise when there is ambiguity in the definition of the relation, as in "is the magnetic south pole to the south of Antarctica?" While I don't suspect these potential confounders would have a very significant effect on the paper's findings, they may explain some portion of the argument switching cases. To what extent are the relations in the World Models dataset unambiguously anti-symmetric? I am curious how the results would change if a third option (e.g., "N/A") were given to the models.

I would not necessarily consider the example in Figure 1 to be one of unfaithfulness to the CoT. In both cases, the answer (no) is faithful with respect to the reasoning in the CoT. The main difference is that the CoTs provide very different arguments.

The authors raise the possibility of another potential confounding factor: sycophancy. While the authors did investigate the possibilty of RLHF-induced sycophancy being the explanation for the same answers being provided to logically opposite questions, they did not test for the presence of sycophancy in the base (pre-RLHF) model. However, the authors did generate data where "yes" and "no" answers are balanced. It would be interesting to compare the unfaithfulness metric on "yes" questions vs "no" questions.

The authors rely on LLMs as a judge for numerous parts of their study, without any validation experiments to ensure the correctness of these LLM-judges (i.e., or "autoraters"). They include a human annotation step in their evaluation pipeline but only for a specific subset of examples, and it is unclear to what extent this step alleviated the stated issues with the LLM judges. It is unclear how accurately these LLM judges perform when rating positive vs negative examples. The "pitfalls" that the authors mention on Line 328 are not elaborated upon.

The authors claim to control for data contamination by experimenting with data that was released after the trainng cutoff of the tested models. However, this does not guarantee that the model has not seen that data, since frontier models are frequently updated, possibly with additional post-training.

The paper is well-written with minimal grammatical or spelling errors.

---

> ### Author Response · Authors · 2025-11-24
>
> Dear Reviewer 7nJy,
>
> Thank you for the thoughtful review. Following your suggestions, in the revised paper we (i) add an IPHR ablation where models can answer "N/A" instead of being forced into Yes/No, and (ii) include a new Validation of LLM Judges appendix with quantitative agreement scores against human annotations. We also address your main concerns below.
>
> We think you raised great theoretical issues that we already largely grappled with during the project, and we hope the clarifications and new experiments make this clearer. If so, we would be very grateful if you could consider increasing your score.
>
> ## Weakness 1 & Question 1: Potential ambiguity in questions
>
> We agree that some relations such as "south of" can be ambiguous or not strictly anti symmetric. To reduce this, **we explicitly built an ambiguity filter for IPHR** (Appendix A.2). First, an autorater flags each question as CLEAR or AMBIGUOUS based on whether multiple interpretations or entity confusions are plausible. Second, for questions marked CLEAR, we check the question and its reversal together and discard any pair where answering Yes to both or No to both would not be logically contradictory. Only pairs that pass both stages are kept.
>
> We also run an ablation study that removes a major ambiguity source by comparing latitudes at similar longitudes and vice versa. This still yields non-zero unfaithfulness for several models.
>
> **Your N/A suggestion.** We implemented your idea and re-ran IPHR for 5 frontier models on all 4,834 pairs with an option to answer "N/A" when neither Yes nor No is clearly justified (**Ablation Study: N/A answers** in Appendix). Most models in this study become modestly less unfaithful, but unfaithfulness does not disappear, and Claude 3.7 Sonnet’s rate even increases slightly. This indicates that forced binarization is not the main driver, and abstention alone does not remove IPHR-style unfaithfulness.
>
> **What we call unfaithfulness in Fig. 1.** For examples like Ajay River / Salar de Arizaro, we agree that each individual CoT is locally consistent with its own answer. Our notion of unfaithfulness is defined at the level of the *pair*: in one direction the model treats cross-continental comparisons as meaningful (implicitly using hemispheres), in the other it effectively declares them meaningless, while never acknowledging the change of standard. The model quietly switches which standard it applies depending on the question order, while never acknowledging this change. We clarify in the revision that IPHR targets this inconsistent choice of reasoning rule across logically opposite prompts, not internal inconsistency within a single CoT.
>
> ## Weakness 2 & Question 3: LLM-as-a-judge validation
>
> In IPHR the judge has a narrow role: given a CoT, it decides whether it supports Yes, No, or Unknown. In the new **Validation of LLM Judges** appendix we report inter-rater reliability between a human rater and the judge (ChatGPT-4o) on 300 responses: **Cohen’s κ = 0.994** on the Yes/No/Unknown decision, showing extremely close alignment with humans.
>
> We also validate the unfaithfulness-pattern classifier derived from our 227-pair case study. Per-category κ values are fair–moderate, but when we collapse Fact Manipulation, Argument Switching, and Other into a single "any pattern" label, we obtain κ ≈ 0.34 with precision ≈ 0.88, recall ≈ 0.98, and F1 ≈ 0.93. Thus the autorater is very reliable as a *binary detector* of whether a pair exhibits any non-trivial unfaithfulness pattern at all, though pattern boundaries are less clear.
>
> For Unfaithful Illogical Shortcuts on the 2024 Putnam subset we use a stricter protocol: an LLM judge first applies eight targeted checks to candidate solutions, and we then manually review **all** responses that pass. Of 17 candidates flagged by the judge, 14 are confirmed by humans (82% precision), indicating good agreement here as well.
>
> ## Question 2: Presence of sycophancy in base models
>
> We share your concern that RLHF induced sycophancy might explain some failures. This motivates our comparison of a base and an instruction-tuned model (Fig. 2). Llama-3.1-70B (base) actually shows *higher* unfaithfulness than Llama-3.3-70B-Instruct, suggesting that RLHF is not the sole cause.
>
> ## Question 4: Control for data contamination
>
> You are right that using problems released after a stated training cutoff cannot fully rule out post training updates, and we now state this explicitly. Our goal in the paper is to **reduce** this risk. The 2024 Putnam exam was released in December 2024, after the November 2024 cutoff stated for the models we test. We ran these experiments in February 2025, using the then current public endpoints. We soften the language in the paper and say that this setup reduces, rather than eliminates, the chance that shortcuts are due to memorised training data, and we flag contamination risk as a limitation.

---

> ### Comment · Reviewer_7nJy · 2025-11-25
>
> I appreciate the author's thoughtful responses to the questions in my review and their proposed revisions to address the concerns.
>
> ### Ambiguity in Questions
>
> The appendix section describing this filter is not referenced until the Limitations section. I suggest that the authors add a description of this filter earlier in the paper, especially since I first had this concern when reading the example in Figure 1. I assume the example in Figure 1 was produced after this filter was applied, which suggests that this filter is clearly not fool-proof and ambiguous examples are still accepted by the filter. The authors write on Line 486-487 that
>  > Despite extensive filtering of ambiguous questions through multiple autorater passes, manual verification, and rigorous criteria (Appendix A.2), subtle prompt ambiguities may remain.
>
> Therefore, the effectiveness of the ambiguity filter is unclear. Was this filter validated with any kind of manual evaluation?
>
> ### Validation of LLM Judges
>
> I appreciate the author's addition of a human validation of the LLM judge's behavior. However, some critical details about the validation are missing. How many human annotators were involved in the study? How were they recruited? How were the options presented to the annotators? Did the humans have any information about the LLM judge's prediction for any given input, or were they blinded to the LLM judge's behavior?
>
> In light of the proposed changes, I will increase the soundness rating in my review. I will await further clarification from the authors before choosing to modify the overall rating.

---

> > ### Author Response · Authors · 2025-11-26
> >
> > Dear Reviewer 7nJy,
> >
> > Thank you again for your thoughtful follow-up and for updating the soundness score. We’ve made **further clarifications and edits to the manuscript** in response to your questions on ambiguity and LLM-judge validation, as detailed below.
> >
> > ## Ambiguity in questions
> >
> > We agree that the ambiguity filter is central to interpreting (and to the soundness of) IPHR, and that it should be mentioned earlier in the paper. In the revision uploaded just now we have:
> >
> > * Mentioned the ambiguity filter when describing IPHR in **Intro** (contributions list),
> > * Introduced the filter at the **start of Section 2**, and
> > * Made the description in **Section 2.1** more explicit, with a clear forward reference to Appendix A.2.
> >
> > Regarding validation: the current ambiguity filter is the result of **five rounds of manual evaluation and refinement** of the pipeline. In each round, three authors inspected a large sample of questions across all IPHR categories (several dozen per category per round), identified recurring ambiguity or questions where answering "false" to both would be fine under some interpretations, and then updated the ambiguity-evaluation prompts and rules to filter those cases out more aggressively. This iterative process is what we refer to in the paper as "multiple autorater passes, manual verification, and rigorous criteria".
> >
> > The Ajay River / Salar de Arizaro example in Figure 1 was indeed generated **after** applying the final ambiguity filter.
> >
> > > Therefore, the effectiveness of the ambiguity filter is unclear.
> >
> > As mentioned, we iteratively refined the ambiguity filter through several rounds of manual inspection, aiming to be as exhaustive as possible. However, in the spirit of being intellectually honest, we still cannot guarantee that all subtle ambiguities are gone, which is why we keep residual ambiguity explicitly flagged as a limitation.
> >
> > Nevertheless, our claims do *not* rely on eliminating every conceivable subtle ambiguity, only on making such cases rare enough that they do not drive the overall unfaithfulness signal. We have updated the Limitations section to convey this message more clearly.
> >
> > ## Validation of LLM judges
> >
> > We have augmented the *Validation of LLM Judges* appendix to clarify who annotated what, how the task was presented, and whether annotators were blinded.
> >
> > * **Who annotated and how they were recruited.**
> >   All human labels in the validation study were provided by **authors from the paper** with experience in LLMs and evaluation, rather than crowd workers. Three authors were involved in total for the manual labelling of the IPHR unfaithfulness patterns and Putnam shortcuts, and one author annotated the 300 IPHR CoT responses used to compute Cohen’s κ with the Yes/No/Unknown judge. We now state this explicitly in the appendix.
> >
> > * **How options and information were presented.**
> >   For the unfaithfulness-pattern study on 227 IPHR pairs, we built a small Streamlit interface that, for each sampled pair, displayed: (i) the two questions (Q1 and Q2) with their correct labels and the model’s correctness rates, (ii) the 20 CoT responses for that pair (10 per direction), and (iii) a dropdown menu of unfaithfulness patterns plus a free-text notes field. As we encountered new recurring behaviors, we expanded the pattern dropdown and refined the taxonomy. For the 300-response Yes/No/Unknown validation, the annotator saw each question, the model's response, and chose one of {Yes, No, Unknown}.
> >
> > * **Blinding with respect to the LLM judge.**
> >   In all of these manual studies, annotators were **blinded to the LLM judge’s outputs**. The 227-pair case study and associated pattern labels were completed **before** we trained or tuned the pattern-level LLM classifier, and the 300 Yes/No/Unknown labels were collected independently of the judge as well. The judge therefore did not influence human labels, and the agreement statistics in the appendix genuinely reflect out-of-sample alignment between human and LLM judgments.
> >
> > We hope these clarifications address the remaining concerns around ambiguity handling and judge validation. We are very grateful for your careful reading and constructive suggestions, which substantially improved both the paper and its presentation.

---

> > > ### Comment · Reviewer_7nJy · 2025-11-26
> > >
> > > ### Ambiguity in questions
> > >
> > > I agree that the claims in the paper do not rely on the elimination of all ambiguity in the questions. However, they do rely on the ambiguity being sufficiently insignificant. But as far as I know, there is no investigation to quantify the degree of the ambiguity. While it seems like the author's process for developing the ambiguity filter was quite thorough, the resulting filter's precision and recall are unknown, as there is no validation of its performance.
> > >
> > > The author's revisions of the _Validation of LLM judges_ section in the appendix have addressed my concerns.
> > >
> > > I have adjusted my score accordingly.

---

> > > > ### Author Response · Authors · 2025-11-28
> > > >
> > > > Dear Reviewer 7nJy,
> > > >
> > > > Thank you again for your careful follow-up, and for pushing us to improve the work. In order to better understand the effectiveness of the ambiguity filter, and the degree of ambiguity in the final dataset, we ran a **new validation study where we: (i) directly measure the ambiguity filter's precision and recall on a fresh set of questions, and (ii) estimate the residual ambiguity in the IPHR dataset**.
> > > >
> > > > We summarize below the methodology and results of this new validation study (also added as appendix in a new revised version of the paper).
> > > >
> > > > ## Setup
> > > >
> > > > We first generated a new set of question pairs across all categories (e.g., book release dates, geographical comparisons, etc), and selected **200 pairs** such that the ambiguity filter has balanced labels over them: 100 pairs of questions marked as CLEAR and 100 pairs marked as AMBIGUOUS. We then randomly sampled **200 pairs** of questions from the existing IPHR dataset, for a **total of 400 pairs of questions in the validation study** (800 individual questions).
> > > >
> > > > Next, **one of the authors of the project labeled each of the 800 individual questions as CLEAR vs AMBIGUOUS**, using a deliberately conservative approach. I.e., if the author was not confident that the question had a unique, unambiguous interpretation under which exactly one of Q(x,y) or Q(y,x) is true, then they would mark it as AMBIGUOUS.
> > > >
> > > > For each question, the annotator was shown the text of the question and the RAG values automatically harvested from online sources, but was **blinded to the ambiguity filter’s label** and to **whether the question came from the new set or from the existing IPHR dataset**.
> > > >
> > > > ## Question-level performance of the ambiguity filter
> > > >
> > > > Restricting to the newly generated questions (N=400 questions, i.e., 200 pairs), we get the following **question-level** confusion matrix (AMBIGUOUS=positive):
> > > >
> > > > > TP = 45, FP = 73, FN = 11, TN = 271
> > > >
> > > > Which yields a **Recall of 0.804** and a **Precision of 0.381** for the LLM-based judge. In other words, **the ambiguity filter is conservative**:
> > > >
> > > >
> > > > - It achieves **high recall** on ambiguous questions (few human-ambiguous questions slip through),
> > > > - At the cost of **lower precision**, i.e., it sometimes discards questions that human annotators would consider clear
> > > >
> > > > As you suggested, this is a tradeoff we are comfortable with here: **false positives only reduce coverage of IPHR**, while false negatives would risk contaminating the unfaithfulness measurement.
> > > >
> > > > ## Residual ambiguity in the existing IPHR dataset
> > > >
> > > > We then asked: *after all filtering, how much ambiguity remains in the datasets we actually use?*
> > > >
> > > > To estimate this, we looked at the **pair-level** human labels on the samples coming from the final datasets (i.e., pairs that the pipeline would actually keep):
> > > >
> > > > * **Existing IPHR dataset**: For the 200 IPHR pairs randomly sampled from the final dataset, **2.0%** were labeled AMBIGUOUS by the human annotator. The 95% CI for the true ambiguous fraction is 0.008–0.050 (N = 200).
> > > >
> > > > * **Freshly generated pairs**: For the 100 newly generated pairs that the ambiguity filter would keep, the human-labeled ambiguous rate was **2.0%**, with a 95% CI of 0.006–0.070 (N = 100).
> > > >
> > > > Pooling these two sources (300 pairs that would be present after filtering), we obtain an overall **residual ambiguity of about 2.0%**, with a 95% **confidence upper bound of 4.5%**. Thus, even under conservative labeling, the fraction of ambiguous pairs in IPHR and in similarly generated data appears to be small.
> > > >
> > > > Taken together with the question-level high recall and these small residual-ambiguity estimates, this supports our claim that the overall pipeline **prioritises removing potentially ambiguous pairs, even at the cost of discarding some clear examples.**
> > > >
> > > > We have added all of this evidence in a new appendix section, "Validation of Ambiguity Filter," and we now cross-reference it from the main body.
> > > >
> > > > We hope this additional validation addresses your remaining concerns about ambiguity handling and the ambiguity filter’s reliability. We really appreciate your suggestions here; they have strengthened the robustness of our claims and clarified how our methodology controls for ambiguity.

---

### Official Review · Reviewer_pcao · 2025-10-31

**Soundness:** 2
**Presentation:** 2
**Contribution:** 2
**Rating:** 4
**Confidence:** 3

**Summary:**

This paper sets out to critique chain-of-thought reasoning in language models by providing a more refined taxonomy of unfaithfulness. They identify particular forms of unfaithfulness like post-hoc rationalizations and illogical shortcuts (perhaps hoping that the reader will be happy to identify these particular subspecies with "unfaithfulness"-full-stop as promised in the title), and they argue that these nuances deserve attention.

To be blunt, while I acknowledge that adding a bit of granularity to the conversation is marginally useful, I’m not particularly moved by this paper. It feels like just another entry in a long line of critiques that don’t really shift the needle; I've had to review three CoT critiques this batch and none of them have made an impression. Yes, the authors have carved out a more detailed definition of what "unfaithfulness" can look like, but they’re not really bringing anything groundbreaking or memorable to the table for me. Let's call this what it is: another minor addition to an already crowded field. I'll lean accept because I don't think it's worth a lengthy debate.

**Strengths:**

- Expands discussion of CoT unfaithfulness with hints of a systematic typology and empirical survey.
- Engages existing literature and situates the problem in real-world prompts (“in the wild”).
- Provides dataset and evaluation scripts for reproducibility.

**Weaknesses:**

- No fundamentally new insight; incremental.
- Analyses largely descriptive: new quantitative or causal tests.
- Offers no solution or mitigation strategy.
- Reads more like a position paper dressed as empirical work.

**Questions:**

- What novelty is claimed beyond earlier works?
- Are the “faithfulness” metrics predictive of task accuracy or just diagnostic labels?
- Could dataset be used to train more faithful models, or is it purely evaluative?

---

> ### Author Response · Authors · 2025-11-24
>
> Dear Reviewer pcao,
>
> Thank you for the candid review and comments. We address, up front, (1) what is new, (2) what our metrics are useful for, and (3) how our benchmarks can be used for mitigation rather than just diagnosis.
>
> **In short:**
>
> 1. Unlike previous CoT faithfulness work that edits completions (e.g. [1]) and/or prompts (e.g. [2]) to inject explicit artificial biases, we use standard factual benchmarks **without adding hand-crafted hints**. Our evaluation **increases the realism of CoT faithfulness evaluation** by uncovering biases that already exist in LLMs, without steering them toward specific answers or modifying rollouts.
> 2. Our faithfulness metrics are *not* just labels on top of accuracy. They separate models with similar accuracy but very different reliability and step consistency.
> 3. We already sketch concrete ways to use IPHR and Unfaithful Illogical Shortcuts to *train* and *monitor* more faithful models in Section 5.1, and there are independent external results (built on our arXiv version) going in this direction.
>
> Below we respond in more detail to your concerns.
>
> ## Weakness 1 & Question 1: Lack of novelty
>
> As mentioned, our work differs fundamentally from previous articles (e.g., [1] and [2]) in that **we do not explicitly add hints/biases, and we do not modify models' rollouts** to evaluate CoT faithfulness. Beyond this key difference, we would like to also mention:
>
> - The breadth of our analysis: we evaluate **15 frontier models from six providers, and two qualitatively different settings** (comparative factual questions and hard math problems).
> - The provided taxonomy of unfaithfulness, and the analysis of prevalence for each of these patterns in frontier models.
>
> ## Weakness 2: Only descriptive analysis; Weakness 3: Lack of mitigation strategies; Question 3: Training more faithful models
>
> We agree that diagnosis must connect to fixes, and Section 5.1 outlines several ways to use our benchmarks. IPHR defines a "consistency with reversal" loss: for each pair of logically opposite questions we can reward correct, consistent answer pairs and penalise inconsistent ones on top of existing supervised or preference based objectives. Template level bias estimates and the probe results in Appendix F suggest simple monitors: if a template or activation pattern indicates strong bias, the system can route that query to a different prompt or model. Because IPHR and Unfaithful Illogical Shortcuts provide structured labels for unfaithfulness, they also serve as held out testbeds for proposed CoT fixing methods.
>
> Understanding "what is going on inside the model" is an important next step for mitigation. As a first move, Appendix F shows that template level answer biases can be predicted from internal activations before the CoT starts, using simple linear probes. Together with the scale and model type trends in our results, this suggests that unfaithfulness is systematic rather than random stylistic noise.
>
> ## Weakness 4: Reads like position paper
>
> While there are conceptual parts in our work, the paper is fundamentally empirical:
>
> * We implement and run a complete evaluation pipeline at scale on comparative factual questions and hard math problems
> * We test 15 frontier models across thousands of questions
> * We provide quantitative results on unfaithfulness rates
> * We carry out case studies that illustrate concrete failure patterns and inform our taxonomy
> * We run ablations on alternative hypotheses and release benchmarks and code for reproducibility
>
> ## Question 2: Faithfulness metrics as diagnostic labels
>
> The faithfulness metrics are **not** predictive of task accuracy. In IPHR, we use answer unbalance as a measure of systematic bias across question pairs, not as a predictor of whether individual answers are correct. A model can get questions wrong due to knowledge gaps while still being balanced (and thus faithful), or it can achieve high accuracy while showing strong template level biases (and thus being unfaithful). As long as responses for a question are not systematically biased toward one answer, our approach does not mark them as unfaithful.
>
> An example where faithfulness differs from accuracy is Sonnet 3.7, which has higher unfaithfulness than Sonnet 3.5 v2 but is generally regarded as the stronger model. Models with similar accuracy can have very different unfaithful pair rates, and some models improve accuracy with little change in unfaithfulness. This shows that our metrics capture a separate "how much can I trust the CoT?" axis.
>
> These metrics inform practical choices: when CoT is used for auditing or explanation based monitoring, one might prefer a model with slightly lower accuracy and much lower unfaithfulness, or use our metrics to flag models or prompt formats that are unsafe for explanation based oversight.
>
> [1] Lanham et al., 2023. Measuring Faithfulness in Chain of Thought Reasoning
>
> [2] Chen et al., 2025. Reasoning Models Don't Always Say What They Think

---

> > ### Comment · Reviewer_pcao · 2025-11-26
> > **With apologies**
> >
> > Thank you for taking the time to respond. I appreciate the spirited defense but I find myself unmoved. My concerns of insufficient conceptual novelty remain, and while it's nice that there are ways to use the metrics for so-and-so, I doubt they're specific to your approach. I will not be changing my score. Perhaps this work will go on arxiv with all the other CoT papers and one day in the future a large science-LLM will deem it worthy.

---

### Official Review · Reviewer_3Ar4 · 2025-11-01

**Soundness:** 3
**Presentation:** 2
**Contribution:** 3
**Rating:** 4
**Confidence:** 4

**Summary:**

This paper focuses on investigating the issue of "Unfaithfulness" in Chain-of-Thought (CoT) reasoning generated by Large Language Models (LLMs). Previous research on CoT unfaithfulness has primarily concentrated on adversarial prompts explicitly designed with human-engineered biases. This paper goes beyond that limitation and systematically demonstrates for the first time that CoT reasoning still exhibits unfaithfulness even when faced with more representative, realistic, and non-adversarial prompts. "Unfaithful" means that the verbalized reasoning steps (CoT) do not accurately reflect the internal computation or logical path the model actually used to reach the final answer. The authors designed a sophisticated multi-stage evaluation methodology to accurately identify and quantify these unfaithful reasoning steps and analyzed their "Criticality" to the final answer, thereby providing a more rigorous diagnostic tool for assessing LLM trustworthiness.

**Strengths:**

1) The paper expands the study of CoT faithfulness from narrow adversarial scenarios to the broader "In the Wild" context, significantly enhancing the practical relevance and application value of this finding.
2) he proposed fine-grained, multi-stage error classification and evaluation framework (as detailed in Appendix K.6, which uses multiple prompts to detect answer correctness, step incorrectness, step unfaithfulness, etc.) is very comprehensive and provides a valuable diagnostic tool for future interpretability research.
3) This work serves as a strong reminder to researchers and developers that a model's decision reliability cannot be judged solely by its verbalized reasoning process. This is crucial for promoting the development of more transparent and trustworthy AI.

**Weaknesses:**

1) The core contribution of the paper lies in shifting to "realistic, non-adversarial prompts," but there is a lack of in-depth justification and formal definition regarding how the chosen benchmarks truly embody "realism." For example, do the three selected datasets (as mentioned in Appendix K.6) merely exclude explicit adversarial bias, or do they statistically approximate the actual queries users submit to search engines or chatbots? The paper should provide a more rigorous analysis (e.g., comparison of perplexity or topic distribution against actual query logs) to enhance the generality and representativeness of the chosen benchmarks. Otherwise, the contribution of the term "realistic" may be diminished.
2) The paper successfully diagnoses the prevalent existence of unfaithfulness but does not offer sufficient evidence to explain its internal mechanisms. Is unfaithfulness an inherent model flaw (e.g., a mismatch between reasoning and conclusion pairings in the training data)? Or is it simply a manifestation of output text fluency exceeding its logical rigour? The paper could attempt to establish causal relationships by systematically varying model architecture (e.g., ablation studies), training data characteristics, or decoding strategies (e.g., temperature, Top-p) to observe trends in unfaithfulness, rather than merely documenting the phenomenon.
3) The paper clearly identifies a critical limitation but lacks adequate discussion or preliminary experiments on how to address or mitigate this limitation.

**Questions:**

1) Regarding the root causes of "unfaithfulness," have the authors tested for any internal diagnostic signals within the model? For instance, do the Attention Maps or the activation patterns of specific layers exhibit recognizable differences when the model generates an unfaithful step?
2) Please provide more detailed information about the benchmark datasets. How were the selected datasets (such as the three mentioned in Appendix K.6) filtered to ensure they were "non-adversarial"? Can quantitative metrics (e.g., perplexity scores of the language model on real user query data) be provided to demonstrate that they are statistically closer to real-world usage scenarios?
3) Please explicitly state whether the primary source for the "Unfaithful" and "Critical" labels was human annotation or LLM meta-evaluation. If human annotation, please provide the corresponding Kappa coefficient or F1 score to measure Inter-Rater Reliability (IRR). If LLM meta-evaluation was used, please explain how you ensured that the meta-evaluator model did not inherit or generate similar unfaithful biases as the model being evaluated.

---

> ### Author Response · Authors · 2025-11-24
>
> Dear Reviewer 3Ar4,
>
> Thank you for the helpful review. Following your suggestion, we added a Validation of LLM Judges study, which we think provides further evidence of the robustness of our claims. We respond to your concerns on realism and mechanisms below. If these experiments and clarifications address your concerns, we would be very grateful if you could consider increasing your score.
>
> ## Weakness 1: Lack of justification for realism
>
> Our main claim about increased realism is that **previous CoT faithfulness work evaluates unfaithfulness by editing prompts or completions, while we do not**. For example, in [1] the authors truncate or corrupt the CoT to test whether the final answer changes, and in [2] they inject biased few-shot examples or misleading context to steer the model toward particular answers.
>
> By "realistic, non adversarial" we mean this move away from explicitly biased setups toward standard benchmarks where any bias comes from the model itself, and we evaluate models **without adding hand crafted hints or editing model outputs**.
>
> Nevertheless, we agree that we do not show that these benchmarks statistically match real query logs. In the revised paper we soften the wording around "realistic" and state more clearly that our contribution is to move from explicitly biased setups to non adversarial, benchmark based evaluations. We already mention in Section 5.1 datasets with multiple answers as future work and now also suggest running our pipelines on real user traces.
>
> ## Weakness 2: Lack of internal evidence; Question 1: Root causes of unfaithfulness
>
> Appendix F already includes an internal diagnostic experiment: we train linear probes on residual activations of Llama 3.3 70B Instruct to predict template level Yes/No bias. Probes at mid layers explain much of the variance in template bias and do so before any visible reasoning starts. This suggests that answer bias is encoded in internal representations ahead of the CoT, although we do not yet provide a circuit level analysis.
>
> We agree that more detailed causal evidence from varying architecture, training data, or decoding is valuable, but in this work **we prioritise breadth: 15 frontier models from six providers in two settings** (comparative factual questions and hard math) under a fixed decoding scheme. Within this setting we already see design relevant trends: reasoning models are more faithful than non thinking variants, and larger models usually show lower unfaithful rates than smaller ones. Section 5.1 flags targeted architectural and training ablations as core future work.
>
> ## Weakness 3: Lack of discussion on mitigations
>
> We agree diagnosis alone is not enough. Section 5.1 outlines two concrete directions as follow up work: (1) a "consistency-with-reversal" regulariser that penalises giving the same answer to logically opposite variants within a template, and (2) "template-gated prompting," where simple monitors, such as activation probes, flag templates with strong answer bias and trigger alternative prompts or templates.
>
> ## Question 2: Selection of datasets
>
> In IPHR we build comparative Yes/No questions from the World Model dataset, with natural phrasings such as "Was A released earlier than B?" over real books, movies, etc.
>
> Appendix A.2 ("Ambiguity evaluation") explains how we keep these pairs non adversarial and unambiguous. We remove obscure or ambiguous entities, require reliable ground truth, and apply a two stage ambiguity check with ChatGPT 4o (single question, then question versus reversal). **Pairs flagged as ambiguous at either stage are discarded**. The resulting questions are hard but not tailored to any particular failure mode.
>
> For Unfaithful Illogical Shortcuts we use three standard math and science benchmarks that were not designed around CoT faithfulness (GSM8K, MATH, and subsets of MMLU), keep the original wording, require a single short final answer, and do not introduce extra biasing instructions.
>
> ## Question 3: Primary source for labels
>
> **We now clarify label sources and validate our LLM judges in a new Appendix section ("Validation of LLM Judges")**
>
> For IPHR, unfaithful pairs come from aggregated LLM-based labels: ChatGPT 4o judges whether each CoT supports Yes, No, or Unknown. This judge achieves a Cohen's  κ=0.994 against human labels on 300 responses. For unfaithfulness patterns, we compare the judge to human tags on 227 pairs: per-category κ is fair–moderate, but an “any-pattern” union reaches κ=0.34 with precision 0.88, recall 0.98, so the judge is reliable as a detector, though pattern boundaries are fuzzy.
>
> For Unfaithful Illogical Shortcuts we rely primarily on LLM meta evaluation, where the model answers targeted questions. We manually vet all the candidates flagged by the judge, and almost all match our intuitive judgement (82% precision).
>
> [1] Lanham et al., 2023. Measuring Faithfulness in Chain of Thought Reasoning
>
> [2] Chen et al., 2025. Reasoning Models Don't Always Say What They Think

---

### Author Response · Authors · 2025-12-03

Dear AC

First of all, we thank you and the reviewers for their time and thoughtful comments. To help you make a decision in these unusual circumstances, we provide an **executive summary of where the paper stands after the (partial) discussion phase**.

We would appreciate you engaging with our rebuttals, given that initial scores do not represent a fair overview of our contributions.

## Executive Summary

* **Original scores:** iSVc **6**, 3Ar4 **4**, 7nJy **4**, pcao **4**.
* After our Nov. 24 rebuttal, 7nJy wrote that some concerns were addressed and **raised their score from 4 → 6**, over half a day before the de-anonymization bug was made public.
* We then ran a **validation experiment** addressing their remaining concern, providing **more evidence for our core claims** and likely supporting a further increase of 7nJy’s score (to **7**) had the discussion continued.
* 3Ar4 did not reply, but we implemented their suggestions (**writing improvements** and a **new validation experiment**), which might have prompted a score increase (to **5–6**).
* iSVc did not reply, but we implemented their suggestions (**writing improvements** and **CIs on all plots**), which might have prompted a score increase (to **7**).
* pcao replied to our rebuttal without engaging our specific points, so it is **unclear to what extent the rebuttal or the work was taken into account.**

## New validation experiments based on the reviewers' feedback

* **Validation study for the LLM judge,** showing strong agreement (Cohen’s κ ≈ 0.994) with humans on IPHR, and high precision/recall for detecting pair-level unfaithfulness (F1 ≈ 0.93).
* **Ambiguity filter validation,** showing high recall for ambiguous questions and low residual ambiguity (≈2% over 400 pairs), indicating that ambiguity is not driving the main results.
* **IPHR N/A ablation study,** showing that letting models answer "N/A" as a third option slightly changes some rates but **does not eliminate** unfaithfulness.
* **Statistical and conceptual clarity,** by adding confidence intervals to all plots in the paper. We also clarified that we study **behavioural** faithfulness, and we rewrote the claims about "realistic" prompts and the limitations section to be more precise.

## Summary of interaction with Reviewer 7nJy

* Describes our work as "an interesting study to quantitatively measure the unfaithfulness of the model's CoT" that "conduct[s] experiments on a wide range of models, including both thinking and non-thinking models," and notes that "the paper is well-written and easy to understand."
* Main concerns: ambiguity in IPHR, LLM-judge validation, and possible sycophancy/contamination.
* We added the LLM judge-validation studies and clarified sycophancy/contamination. The reviewer acknowledged that these addressed their concerns and **raised the score from 4 → 6**.
* After this, we added an ambiguity filter validation experiment targeting their last open issue.

## Summary of interaction with Reviewer 3Ar4

* Says we "expand the study of CoT faithfulness ... to the broader 'In the Wild' context" and that our framework is "very comprehensive" and "a valuable diagnostic tool."
* Main concerns: how "realistic" the prompts are, limited mechanism analysis, and limited mitigation discussion.
* We clarified the notion of "non-adversarial benchmark-based" prompts, internal evidence, proposed mitigations and selection of data/labels.
* The LLM judge-validation experiment mentioned above also addresses one of Reviewer 3Ar4's concerns.

## Summary of interaction with Reviewer iSVc

* Highlights a "systematic analysis of the faithfulness problem in CoT reasoning" and experiments that "offer valuable insights and guidance for future large language model research."
* Main concerns: limited internal mechanism analysis, missing CIs, task coverage, and clarity on what "faithfulness" means.
* We added CIs to all plots, clarified task scope, and stated clearly that we use a behavioural notion of faithfulness with explicit limitations.

## Summary of interaction with Reviewer pcao

* Acknowledges that we "expand discussion of CoT unfaithfulness," and that our work "situates the problem in real-world prompts" and "provides dataset and evaluation scripts for reproducibility."
* Main concern: they see no fundamentally new insight, viewing the analysis as largely descriptive.
* We clarified that prior CoT faithfulness work typically **edits prompts or completions to inject artificial biases**, whereas we use a **non-adversarial, benchmark-based setting without such edits**, and that we implement and run a concrete evaluation pipeline across **15 frontier models.**
* Reviewer pcao's reply does not address our claims for novelty. Instead, it just mentions that they find themselves "unmoved" and that "maybe one day a large science-LLM will deem this work worthy," which we find not very constructive.

Again, thank you for your time and for considering this context as you make your decision.

---

### Meta-Review · Area_Chair_PwAM · 2026-01-07

**Summary:**

This paper presents a systematic empirical investigation of unfaithfulness in Chain-of-Thought reasoning, demonstrating that unfaithful reasoning persists even under non-adversarial, benchmark-based prompts. Reviewers broadly agree that the study is careful, well-executed, and valuable as a diagnostic contribution. The rebuttal addressed several substantive concerns, including validation of LLM judges, ambiguity filtering, and statistical reporting, strengthening the technical credibility of the results.

**Reviewer Concerns:**

Addressed concerns -

* Validity of LLM-as-a-judge evaluation
* Ambiguity in anti-symmetric relations
* Breadth of model evaluation

Outstanding concerns -

* The work does not meaningfully probe internal model representations (e.g., attention, activations, logits) to explain why unfaithfulness arises.
* The paper diagnoses the problem but provides little empirical evidence on how unfaithfulness could be reduced or controlled in practice.
* Despite careful analysis, the paper does not introduce a fundamentally new conceptual framework.

**Reviewer Scores:**

I dont think there would have any change.

---

### Decision · Program_Chairs · 2026-01-26

Reject